# Surgical Management of Tracheal Invasion by Well-Differentiated Thyroid Cancer

**DOI:** 10.3390/cancers13040797

**Published:** 2021-02-14

**Authors:** Fumihiko Matsumoto, Katsuhisa Ikeda

**Affiliations:** Department of Otorhinolaryngology, Juntendo University Faculty of Medicine, Tokyo 113-8431, Japan; ike@juntendo.ac.jp

**Keywords:** well-differentiated thyroid carcinoma, trachea, invasion, shaving, window resection, sleeve resection, circumferential resection, end-to-end anastomosis

## Abstract

**Simple Summary:**

Tracheal invasion is a poor prognostic factor in well-differentiated thyroid cancer. Appropriate resection can improve the prognosis and maintain the patient’s quality of life. Shaving resection for superficial tracheal invasion is minimally invasive because it does not involve the tracheal lumen, despite the problematic risk of local recurrence. Window resection for tracheal mucosal and luminal invasion provides good tumor control and does not cause postoperative airway obstruction; however, the need for surgical closure of the tracheocutaneous fistula is a disadvantage of this method. Circumferential (sleeve) resection and end-to-end anastomosis are highly curative, but the risk of fatal complications, such as anastomosis dehiscence, is a concern.

**Abstract:**

Well-differentiated thyroid carcinoma (WDTC) is a slow-growing cancer with a good prognosis, but may show extraglandular progression involving the invasion of tumor-adjacent tissues, such as the trachea, esophagus, and recurrent laryngeal nerve. Tracheal invasion by WDTC is infrequent. Since this condition is rare, relevant high-level evidence about it is lacking. Tracheal invasion by a WDTC has a negative impact on survival, with intraluminal tumor development constituting a worse prognostic factor than superficial tracheal invasion. In WDTC, curative resection is often feasible with a small safety margin, and complete resection can ensure a good prognosis. Despite its resectability, accurate knowledge of the tracheal and peritracheal anatomy and proper selection of surgical techniques are essential for complete resection. However, there is no standard guideline on surgical indications and the recommended procedure in trachea-invading WDTC. This review discusses the indications for radical resection and the three currently available major resection methods: shaving, window resection, and sleeve resection with end-to-end anastomosis. The review shows that the decision for radical resection should be based on the patient’s general condition, tumor status, expected survival duration, and the treating facility’s strengths and weaknesses.

## 1. Introduction

Well-differentiated thyroid carcinoma (WDTC) is a slow-growing cancer that has a good prognosis. The trachea has posterior anatomical proximity to the thyroid gland and is affected by extraglandular extension of thyroid cancer. Tracheal invasion by WDTC is infrequent, exemplified by a rate of 3.4–13% [1,2,3,4,5]. Since this condition is rare, relevant high-level evidence about it is lacking. In tracheal invasion by thyroid cancer, the symptoms range from none to fatal, such as airway obstruction and bleeding. McCarty et al. [6] reported that 22% of 40 patients with obvious tracheal invasion had hoarseness, 11% had hemoptysis, and 5% had respiratory distress. In general, tracheal tumor invasion indicates higher biological tumor aggressiveness compared to tumors limited to the thyroid gland [7]. McCaffrey et al. [8] and Ito et al. [9] found that invasion of the aerodigestive tract was an important negative prognostic factor. Thus, data indicate that tracheal invasion decreases long-term survival; intraluminal tumor development constitutes a worse prognostic factor than superficial tracheal invasion [10,11]. However, in WDTC, even with tracheal invasion, curative resection is often possible with a large margin of safety, and complete resection can ensure a good prognosis. Despite tracheal invasion, WDTC has an excellent prognosis, with more than 90% of patients surviving to 10 years after complete resection [12,13,14]. A far advanced or rapidly progressing tumor may be unresectable, and only tracheostomy or stenting can be undertaken. Even in resectable WDTC, accurate knowledge of the anatomy of the trachea and its adjacent tissues, as well as the proper selection of surgical techniques, is necessary for complete resection. Current guidelines do not provide specific guidance on the resection of tracheal invasion by WDTC. The 2015 American Thyroid Association Management Guidelines for Adult Patients with Thyroid Nodules and Differentiated Thyroid Cancer indicated that the decision for surgical treatment should be based on the feasibility of complete resection and the resultant functional impairment following resection; however, the surgical indications and the recommended procedure in trachea-invasive WDTC were not specified [15].

On a case-by-case basis, the possibility of resection, expected postoperative functional disability, and patient’s background (e.g., general condition and expected prognosis) should be considered before deciding on surgical intervention. This review discusses the indications for radical resection and the three major resection methods currently available: shaving, window resection, and sleeve resection with end-to-end anastomosis (Table 1).

## 2. Anatomy of the Trachea

The trachea extends from the inferior edge of the cricoid cartilage to the tracheal carina. The anatomy of the trachea has been described previously [16], and the length of the adult trachea ranges from 10 to 13 cm, with an average length of 11.8 cm. Each tracheal ring is C-shaped with a membranous posterior portion and is approximately 0.5 cm long; there are 18–22 tracheal rings in total. The inner diameter of the trachea is 2.3 cm and 1.8 cm laterally and posteriorly, respectively. If the neck is flexed, the trachea may be completely hidden within the mediastinum. Conversely, if the neck is extended backward, the trachea will be elongated and becomes visible in the neck. These observations are especially important when associated with tracheal resection. The blood supply is segmented and laterally approaches the trachea. The upper trachea is perfused by the inferior thyroid artery, whereas the lower trachea is perfused by the bronchial arteries, which receive blood flow from the subclavian, internal mammary, innominate, internal thoracic, and the superior intercostal arteries [17].

## 3. Evaluation of Tracheal Invasion

Preoperative evaluation of tracheal invasion is important in determining the surgical procedure. Tracheal invasion may reach the adventitia, intra-cartilage, submucosa, or lumen. Therefore, it is important to evaluate not only the presence but also the degree of invasion. Modalities for evaluation of tracheal invasion include computed tomography (CT), magnetic resonance imaging (MRI), bronchoscopy, and endobronchial ultrasound (EBUS). In cases where ultrasonography (US) or CT show the extrathyroidal extension and tracheal invasion is suspected, further evaluation is recommended.

### 3.1. US

US is the first examination to be performed because it is minimally invasive, relatively inexpensive, well-tolerated, and does not require contrast agents. Yamamura et al. [18] evaluated 24 patients with differentiated thyroid cancer invading the airway who underwent various types of tracheal resection. Preoperative US findings in these patients were compared with postoperative histologic results of the resected specimens to confirm the relation between the preoperative and postoperative diagnoses and determine the indications for operative procedures. Preoperative US revealed cancer invasion to the adventitia in 2 cases, the intra-cartilage space in 13 cases, and the tracheal mucosa in 9 cases. Histologically, cancer invasion to the adventitia was confirmed in 4 cases, the intra-cartilage space in 10 cases, and the mucosa in 10 cases. Among the 13 instances of cancer invasion to the intra-cartilage revealed by US, overdiagnosis occurred in 3 cases and underdiagnosis in 1 case. The actual overall diagnostic rate by US was 83.3%. Preoperative US findings of the trachea were consistent with the histologic findings of the resected specimens. Tomoda et al. [19] reported that the accuracy of US for diagnosing tracheal invasion was 93%, and the negative predictive value was 99%. In contrast, the positive predictive value was low, at 25%. However, another study showed that the sensitivity of ultrasound to invasion to the trachea and esophagus was 42.9% and 28.6%, respectively [20]. These different findings are due to varied US performance and diagnostic criteria for tracheal invasion. US has several disadvantages in the diagnosis of tracheal invasion. It is not reliable for tumors larger than 4 cm in diameter or tumors with significant calcification. In addition, its diagnostic ability is quite dependent on the surgeon’s skill [20].

### 3.2. CT

The American Head and Neck Society consensus statement indicates [21] that CT is acceptable for assessing the status of tracheal invasion and is superior to US. However, there are few reports evaluating tracheal invasion in differentiated thyroid cancer using CT. A study by Seo et al. [22] defined tracheal invasion as at least one of the following CT criteria: tumor in contact with 180° or more of the tracheal circumference; deformity of the tracheal lumen at the level of the mass; and focal irregularity, thickening, or bulging in the mucosal portion adjacent to the mass. The mean sensitivity, specificity, and accuracy of CT for tracheal invasion were 59.1%, 91.4%, and 83.2%, respectively.

### 3.3. MRI

Similar to CT and US, MRI has been shown to be useful in evaluating tracheal invasion in several reports. Wang et al. [23] evaluated the efficacy of MRI for the diagnosis of tracheal invasion. MRI was performed on the normal trachea of one cadaver and 30 healthy subjects as reference standards. Then, MRI findings of 67 patients with thyroid carcinoma were reviewed and correlated with surgical and pathologic findings. MRI characteristics indicative of tracheal invasion included soft tissue signal in the tracheal cartilage and lumen, and the degree of tumor circumference around the trachea was significantly related to surgical and pathologic findings. The highest accuracy (90%) for determining tracheal invasion was achieved using a combination of findings. Although an intraluminal mass invariably reflected deep tracheal invasion, a soft tissue signal in the cartilage rarely indicated actual cartilage invasion but rather tumor extension between the cartilaginous rings. The authors concluded that the ability of MRI to predict the precise depth of tumor invasion to the trachea was unreliable. However, the presence or absence of tracheal invasion was accurately diagnosed with this procedure. Takashima et al. [24] reviewed surgical and MRI findings in 50 patients with differentiated thyroid carcinoma. The degrees of circumference of tumor encroachment to the organs were measured, and the measurements and morphologic diagnosis of tumor invasion made by a head and neck radiologist were compared with surgical and pathologic findings. With a threshold of 135°or more of tumor circumference, they obtained the highest accuracy (86%), with 71% sensitivity and 92% specificity for tracheal invasion. These findings suggest that this simple method of measuring tumor circumference will be clinically beneficial. The diagnostic accuracy of MRI in this study was 73%, which is lower than that of US. Notably, the early stage of tracheal invasion is more difficult to identify by MRI than by US because of the thin adventitia and degradation of the image by motion artifacts. In contrast, the positive predictive value of MRI was 71%, which is higher than that of US [19].

### 3.4. Bronchoscopy

Bronchoscopy is useful for observing infiltration into the tracheal lumen and changes in the tracheal mucosa. Although it is invasive, laryngeal fiber can be used as a substitute, and it is a familiar examination for otorhinolaryngologists. In esophageal cancer, the usefulness of bronchoscopy for the evaluation of tracheal invasion has been demonstrated [25,26,27,28]. However, in esophageal cancer, the tumor invades the trachea from the posterior wall through the membranous area without cartilage. Thyroid cancer usually invades the trachea between the cartilage rings. Therefore, it is problematic to use the findings of esophageal cancer for tracheal invasion in thyroid cancer. Koike et al. [29] compared the bronchoscopy findings (localized mucosal redness, telangiectasia, mucosal elevation, mucosal edema, and mucosal erosion) with pathological results in 30 patients who underwent curative resections. Of the 18 patients without localized mucosal changes, shaving of the laryngotracheal wall was performed in 4 patients because of laryngotracheal invasion identified during surgery. Shaving of the laryngotracheal wall was performed successfully in terms of obtaining a caner free margin. Twelve patients with localized mucosal redness required extensive resections. Other mucosal changes were found depending on the depth of cancer invasion.

### 3.5. EBUS

EBUS is excellent for assessing the presence and degree of the tracheal invasion by visualizing the layered structure of the tracheal walls from the tracheal lumen’s side. Wakamatsu et al. [30] compared CT, MRI, and EBUS to evaluate tracheal invasion by thyroid cancer. In cases with suspected contact between the tumor and tracheobronchial wall, CT, MRI, and EBUS indicated deformity of the tracheobronchial wall due to the adjacent mass. The sensitivity and specificity of CT, MRI, and EBUS for invasion were 59 and 56, 75 and 73, and 92 and 83%, respectively. The accuracy of EBUS was significantly greater than that of CT. The authors concluded that EUBS is the most useful technique for determining the depth and extent of tumor invasion into the airway wall. However, EBUS has several disadvantages, including invasiveness, relatively high cost, and limited utility in evaluating tumors in the upper parts of the thyroid lobe [22].

Several studies have compared methods of evaluating tracheal invasion. According to Wakamatsu et al. [30], MRI and EUBS are superior in diagnosing tracheal invasion of thyroid cancer, and EBUS is especially beneficial because it allows visualization of the invaded tracheal layer. Although the authors did not use multi-slice and three-dimensional CT, they reported a lower CT accuracy than in other reports, meaning it is out of reference. As mentioned earlier, EBUS is not easy to perform due to its invasiveness and complicated examination. CT and US are easy to perform and are often performed at the diagnosis stage of thyroid tumor. If tracheal invasion is suspected, a bronchoscopy should be considered, or an otorhinolaryngologist should check the lumen with a laryngeal fiber. Finally, it is necessary to diagnose tracheal invasion of thyroid cancer using tests that can be performed irrespective of the facility and the patient’s condition. Despite the usefulness of the aforementioned tests, diagnosis is difficult because it depends largely on the diagnostic criteria and the diagnostic ability of the individual.

## 4. Surgical Techniques

### 4.1. Tangential Tumor Excision/Shaving 

Tangential tumor excision/shaving is used for thyroid tumors that superficially invade the trachea. This method involves sharp resection of the invading thyroid tumors, which are difficult to remove by blunt dissection, from the tracheal cartilage or membrane using a scalpel. Complete macroscopic resection should be performed, although the resection should not extend into the tracheal lumen. Occasionally, electrocauterization may be performed intraoperatively if a microscopic residual tumor can be identified.

Tsukahara et al. reported that in cases where macroscopic examination reveals that the tumor invasion has not reached the tracheal mucosal surface, a shaving tracheal surface resection achieved local control in 21 (95%) of their 22 patients. Thus, the authors concluded that if the tumor did not show macroscopically evident extension to the tracheal mucosal surface, local control by tracheal shaving would be feasible [31]. The fact that the resection does not extend into the tracheal lumen is a major advantage, as the patient does not suffer from serious postoperative complications that reduce their quality of life, such as the maintenance of a tracheocutaneous fistula, tracheal dehiscence, and infection. Ito et al. scraped the tracheal cartilage and ligament between the cartilage after carcinoma resection and confirmed that these areas were carcinoma-negative by prompt pathological examination using a frozen section in 13 patients. None of these patients showed recurrence. They suggested performing shaving resections with intraoperative pathological confirmation using a frozen section of scraped materials from the tracheal wall surface, when possible. In cases with suspected remnants of carcinoma, further shaving should be performed [32]. However, some authors noted that it is difficult to confirm negative margins intraoperatively using frozen sections [33]. Therefore, the possibility of microscopic tumor remnants exists. Studies that reported the benefit of shaving for superficial tracheal invasion are based on the theory that the microscopic residual tumor does not influence local recurrence or prognosis. This is a major point for discussing the validity of this surgical procedure, and opposing opinions have been mentioned in published reports. In some studies, a statistically significant survival adverse effect of microscopic residual tumor on the trachea or shaving off of the thyroid tumor from the trachea was demonstrated when compared to a clear margin on final pathology [6,34,35,36]. However, several retrospective reviews have failed to show a survival advantage of resections with negative margins over resections with microscopically positive margins [8,32,37,38,39,40,41,42].

One of the reasons underlying the utility of the shaving technique for superficial tumor invasion of the trachea in thyroid cancer is that superficial invasion is difficult to define and diagnose. This is supported by the classification of tracheal invasion developed by Shin et al. [43], which suggests that thyroid cancer in contact with the tracheal wall may extend to the submucosa through the soft tissue between the tracheal cartilages without invading the tracheal cartilage. In such cases, despite apparent superficial invasion, resection of the tracheal lumen is necessary. Second, it is speculated that there may be differences in the quality of macroscopic tumor resection from across studies, which queries the quality of the shaving technique. In case the tumor has not spread to the luminal mucosa or submucosa, shaving would be useful in ensuring low morbidity for tracheal-invasive WDTC.

In a recent report, Kim et al. [5] noted that the local recurrence rate in the shave excision group was not significantly different from that in the other groups with more radical excision, although half of the patients showed a positive surgical margin. Furthermore, the disease-specific survival rate (DSS) in the shave excision group was 100%, even at the 10-year follow-up. The aforementioned finding is further supported by Wang et al. [44], who considered the 5-year DSS of patients based on resection status (R0/R1/R2), and the DSS decreased incrementally across the three resection statuses (94.4%, 87.6%, and 67.9%; *p* = 0.030). It appears that, although the presence of microscopic residual disease did not affect recurrence-free survival, microscopic tumor remnants had a detrimental impact on survival, which was associated with a significant difference in DSS between patients with R0 and R2 statuses. However, the data in their study did not demonstrate a statistically significant difference in survival between the R0 and R1 groups.

In the 2016 study by Su et al. [45], no statistically significant difference was found in overall survival or disease-free survival (DFS) between resection groups with negative margins and groups with microscopically positive margins or gross disease in WDTC. However, nearly all patients received postoperative external beam radiotherapy (EBRT) and thyroid-stimulating hormone (TSH) suppression. In some of the studies that reported the benefit of shaving excision for superficial tracheal invasion, the patients received similar postoperative EBRT as that described by Su et al. [38,45]. In this regard, Ito et al. [46] described the efficacy of postoperative EBRT. Sixty-five PTC patients who received shaving or laminated resection for minimally invasive tumor to trachea were classified as the suspicious group or diagnosed as being carcinoma-positive at the resected tracheal surface. Nineteen patients underwent postoperative EBRT. However, the local DFS rate of patients who underwent EBRT did not differ from that of those who did not undergo EBRT. Similarly, radioiodine therapy did not improve the local DFS rate of patients. Therefore, there is no highly validated evidence that the aforementioned treatment improves local prognosis in patients with minimal tracheal invasion. The efficacy of EBRT for microscopically residual tumors after shaving is uncertain and may not be a major consideration when comparing surgical outcomes. Future prospective studies of surgical methods with uniform postoperative treatment should be conducted to further elucidate this concern.

### 4.2. Full-Layer Resection

There are two major surgical techniques for the resection of intraluminal tracheal invasion by WDTC: window resection and circumferential resection (sleeve resection) followed by end-to-end anastomosis. Each surgical technique has its advantages and disadvantages as well as limitations with regard to the indications.

#### 4.2.1. Window Resection

Window resection is a technique wherein the tracheal wall invaded by thyroid cancer is resected in all layers with a safety margin. In cases in which the resection range is limited or only the posterior membranous tracheal wall is preserved, it can be sutured primarily without requiring a tracheostomy or tracheocutaneous fistula. If the excision is larger, a tracheocutaneous fistula is constructed. Several operations will be required to achieve closure of the fistula. The advantages of a window resection include a stable postoperative airway, less invasive surgery, low rate of surgical complications, and avoidance of postoperative fixation of the neck region [47]. This procedure is particularly useful in cases where the range of invasion is limited to half or less than half of the circumference of the tracheal wall [48,49]. However, Ebihara et al. [47] concluded that, in many cases of invasion of thyroid carcinoma into the trachea, lesions are unevenly distributed on either the left or right side, and there is an assumption that only a few cases require circumferential resection. The window method can ensure an adequate resection range and does not require low-level preventive tracheotomy, which is only considered in special conditions, such as bilateral vocal cord paralysis, anastomotic dehiscence, or dyspnea [50]. Ozaki et al. [51] reported that window resection led to a high frequency of positive margins, as histologic examination of the circumferential spread showed that invasion on the mucosal side exceeded invasion on the adventitial side. However, in a study by Ebihara et al. [47], no significant differences in treatment outcomes were observed between the stump-positive group and the stump-negative group in the patients who underwent window resection for tracheal invasion by thyroid carcinoma.

##### a: Primary Direct Closure and No Tracheostomy

If the tracheal resection is not particularly extensive, primary suture is possible and tracheostomy or tracheocutaneous fistula creation are not necessary. Even if a large resection is performed, if only the posterior membranous tracheal wall is preserved, it can still be sutured, as reported by Tsukahara et al. [52]. They performed tracheal resection with primary anastomosis, preserving a posterior membranous portion of the trachea in 12 patients. Although six of these showed various complications, such as bilateral recurrent laryngeal nerve paralysis, subglottic stenosis requiring reintubation, and esophageal fistula, no patients displayed necrosis or serious suture failure, and no local recurrence was identified. If primary suture of the tracheal defect is achieved, the lumen of the trachea would be covered with normal mucosa. This would be the ideal airway, which would not raise concerns about adhesion of crusts.

##### b: Window Resection with Tracheostomy or Tracheocutaneous Fistula

Once a tracheocutaneous fistula is constructed, primary closure of the fistula may be difficult in some cases, requiring reconstruction using various techniques. Window resection often requires multi-staged surgeries for reconstruction, which prolongs hospital stay and can adversely affect the quality of life. In addition, the inner surface of the reconstructed trachea is covered by skin and lacks mucosal epithelium and mucus secretion. A dry surface and squamous debris may cause chronic cough, discomfort, and obstruction of the respiratory tract. Small tracheal defects, up to a length equivalent to seven rings, can be closed nonproblematically using local skin flaps [47]. A much smaller fistula may close spontaneously merely by removing the tracheal canula. Ito et al. [46] reported the results of closure with a local flap wherein they enrolled 109 patients who underwent window resection of the trachea for PTC. The stoma was closed 3–6 months later using a local skin flap that surrounded the stoma in the second-stage surgery. The tracheocutaneous fistula was closed secondarily in 78 patients, and it closed spontaneously in 13 patients. The remaining 18 patients did not achieve fistula closure for various reasons, such as a poor general condition, permanent bilateral vocal cord paralysis, and loss to follow-up. None of the 109 patients showed any serious complications. When the resection range is wide, as in cases of defects larger than half of the tracheal circumference or with tracheal defects equivalent in diameter to up to 10 rings of the trachea, grafting of hard tissues or use of graft materials are required to avoid a collapse of the soft-tissue roof, to achieve an intact luminal lining, and to restore tracheal function for clearing secretions [53]. Yu et al. [53] highlighted the need for immediate reconstruction of oncological tracheal defects with well-vascularized tissue, for example, via a microvascular free tissue transfer. However, immediate reconstruction of a tracheal window defect is often avoided due to technical difficulties and complexity, and the defect is instead transformed into a tracheostoma and closed secondarily, if possible. Several approaches to tracheal reconstruction, including autologous rib cartilage grafts [36], composite nasal septal grafts [54], pedicled sternocleidomastoid clavicular periosteocutaneous flap [55], posterior tibial artery perforator flap [56], free radial forearm flap combined with biodegradable mesh suspension or costal cartilage [57,58], latissimus dorsi musculocutaneous flap [59], and free radial forearm flap combined with titanium mesh or rib cartilage [53,57,60], have been employed. However, each method has its shortcomings. Thus, the decision on the reconstruction method should be based on the size and location of the tracheal defect and the patient’s condition.

#### 4.2.2. Circumferential (Sleeve) Resection and End-to-End Anastomosis

Circumferential resection is performed when the invading thyroid carcinoma extends into the tracheal mucosa or lumen. After resection, end-to-end anastomosis is considered the ideal airway reconstruction technique because airway reconstruction can be completed in one stage, and the airway lumen would be covered by normal tracheal mucosa [61,62,63]. Tracheal resection and primary reconstruction are feasible surgical procedures for patients with thyroid cancer infiltrating the upper aerodigestive tract, with good clinical outcomes.

After standard thyroidectomy, the tracheal segments showing invasion, including the membranous portion, are resected circumferentially while observing the tracheal lumen. The advantages of this approach include a full-thickness specimen of the trachea for the precise determination of invasion depth and margin status and visualization of the trachea with the residual tumor [64]. The resection line is made at the point of one tracheal ring proximal and distal to the invasion margin. The decision as to how many tracheal rings are to be resected is made by the surgeons based on the gross findings of adventitial involvement during operation [48,51]. Tracheal and esophageal clearance are fully released. Before suturing, reintubation was performed through the oral end. After intubation, an end-to-end anastomosis is performed. The sutures are placed circumferentially through the full thickness of the trachea [48,65]. Musholt et al. [66] described the technique to reinforce the anastomosis and protect the anastomotic site and adjacent large vessels with the sternocleidomastoid muscle flap. A tracheostomy should be avoided after end-to-end anastomosis as tracheal secretions contaminate the surgical field and predispose to wound infection [67]. Thus, it is necessary to manage possible bilateral recurrent nerve paralysis, particularly that of a unilateral nerve that is preoperatively paralyzed by disease. It is critically important to release an intact recurrent laryngeal nerve from the trachea and the membranous esophagus to avoid injury during tracheal resection and reconstruction. Tracheostomy is only considered in special conditions such as bilateral vocal cord paralysis, anastomotic dehiscence, or dyspnea [50].

Anastomotic dehiscence is one of the most feared complications as it is a life-threatening condition that can occur in all types of tracheal resections. In a review article, postoperative complication rates after tracheal sleeve resection with end-to-end anastomosis ranged from 15% to 39%. Postoperative mortality is approximately 1.2% [68]. Older reports show a higher incidence of complications, ranging from 4% to 14%, with a mortality rate of 7.8% in a large series [69]. More recent reports indicate that the results have improved, although serious complications are inevitable. Nakao et al. [70] demonstrated that, in 40 patients who underwent end-to-end anastomosis of the trachea, insufficiency of the anastomosis occurred in four patients. Two of the four patients died of massive bleeding from the large artery, and one patient died of mediastinitis caused by insufficiency. Wright et al. [71] performed end-to-end tracheal resection in 901 patients with various diseases. Anastomotic complications occurred in 81 patients (9%). Eleven patients (1%) died postoperatively: six due to anastomotic complications and five due to other causes. Stepwise multivariable analysis revealed the following predictors of anastomotic complications: reoperation, diabetes, lengthy resections, laryngotracheal resections, young age (pediatric patients), and the need for tracheostomy before operation. Shenoy et al. [72] reported that one of seven patients who underwent circumferential resection died of septicemia. Therefore, insufficiency of the tracheal anastomosis should be avoided. However, there are no standard procedures or guidelines for tracheal resection in the literature. The surgeon must preserve the blood supply, which originates from the inferior thyroid artery and passes laterally along the tracheal wall, and must avoid ischemia of the mobilized tracheal segment. Therefore, mobilization of the cervical and mediastinal portions of the trachea is best accomplished with blunt dissection along the anterior wall with minimal lateral dissection. Sutures are placed in the airway with knots placed outside the lumen to minimize the intraluminal reaction and the associated risk of tracheal stenosis [73]. The first technique to prevent anastomosis dehiscence is to flex the chin to the chest position with two interrupted sutures for 5 to 15 days. However, the use of this technique is controversial. At present, some authors continue to advocate the need for postoperative fixation of the neck [69,74], whereas, in some highly experienced centers, chin stitches are applied only in cases of resections larger than 4 cm and re-resections [75]. Tran et al. [63] suggested that it is important to counsel patients to avoid neck extension for the first week after surgery instead of the routine use of the abovementioned stitch. Second, an important aspect to consider with this technique is the postoperative intubation of the tracheal tube. There is no consensus on the timing of tracheal tube removal, and this should be decided according to the experience of each center. Nevertheless, early extubation has been suggested to limit the effects of positive ventilation on the tracheal suture line [71,76].

Finally, it is important to avoid anastomosis dehiscence and suture with less tension. One technique for reducing anastomotic tension is suprahyoid laryngeal release. This technique involves transection of the mylohyoid, geniohyoid, and genioglossus muscles and the lesser cornua of the hyoid bone resulting in a laryngeal drop of 2–3 cm [48,77]. Suprahyoid muscle release is often associated with temporary dysphagia and aspiration; therefore, patients with short tracheal resections generally do not require suprahyoid releases. One of the most important risk factors for anastomotic dehiscence is the length of the resected tracheal segment, which is directly correlated with the tension over the suture line. Tracheal resections longer than 4 cm are associated with a remarkable increase in the rate of anastomotic dehiscence. Thus, in these cases, it is important to dissect the suprahyoid muscles from the hyoid bone to release the larynx to perform end-to-end tension-free anastomosis [78]. Chen et al. [74] noted that end-to-end anastomoses might not require suprahyoid muscle release when <8 tracheal rings have been resected. Other methods include median sternotomy, which is required for mediastinal releasing procedures that mobilize the distal trachea. Sharp dissection is carried downward to the carina and along the proximal major bronchi to free the mediastinal air passages of the surrounding tissues as well as pleural and pericardial attachments [79]. A method for reducing tension in anastomotic sites by pulling the distal segment of the trachea using sternal flaps of the sternocleidomastoid muscle has also been described [80]. The decision to perform these maneuvers should be based on the extent of the resection, the degree of tension in the sutures, the length of the patient’s neck, and the degree of scarring in the neck. Previous studies demonstrated that this technique is applicable for tracheal resection of up to seven rings or a major axis of 5–6 cm [38]. Ozaki et al. [51] reported nine rings or up to 7 cm, and Nakao et al. reported that up to 11 rings were resectable.

Thyroidectomy associated with tracheal resection with end-to-end anastomosis is universally accepted to achieve local tumor control. Complete resection of local disease has been reported to provide the longest survival [11,37,48,65,67,71,72,74,81,82]. Ozaki et al. [51] compared the extent of invasion of the tracheal wall on the adventitial and mucosal sides based on histologic examination. The invasion on the mucosal side never exceeded that on the adventitial side. In contrast, an examination of circumferential spread showed invasion on the mucosal side that exceeded that on the adventitial side. Therefore, the authors suggested that when the extent of invasion of the adventitia is considered, tumoral tissue may remain on the mucosal side of the trachea if a partial window resection is performed. Thus, whenever feasible, circumferential sleeve resection should be performed in patients with thyroid carcinoma that invades the trachea.

## 5. Conclusions

There is a general consensus that the order of curability of radical surgical treatment for tracheal invasion is sleeve > window > shaving resection. However, shaving is the least invasive surgery. In the case of superficial invasion of the trachea up to the perichondrium or cartilage, there is consensus that shaving is generally sufficient if the preoperative evaluation is accurate, although it is sometimes difficult to determine the degree of invasion. In a much deeper invasion, many reports indicate that shaving may increase the risk of local recurrence because of microscopic tumor remnants; therefore, shaving may not be sufficient. Moreover, there is consensus that a full-laminar resection is necessary if the disease extends into the lumen or tracheal mucosa. In such cases, there are two possible resection methods: window resection and circumferential resection. Notably, the perioperative risk of sleeve resection is higher than that of window resection, and window resection is not less invasive due to the need for another surgery to close the tracheocutaneous fistula and the possibility of stenosis. However, apart from these complications, window resection is relatively safe, which is a great advantage. Nonetheless, the evidence is inconclusive and requires prospective review. The decision will be based on the patient’s general condition, tumor status, expected duration of prognosis, and the facility’s strengths and weaknesses (Figure 1). In cases of advanced disease or poor patient condition that makes radical resection intolerable, a less invasive approach should be provided. For this reason, accurate preoperative workup is necessary for choosing the suitable surgical procedure for patients with suspected tracheal invasion. CT, US, and bronchoscopy are essential tests for preoperative evaluation. In addition, MRI and EBUS should be considered depending on the situation (Figure 1).

## Figures and Tables

**Figure 1 cancers-13-00797-f001:**
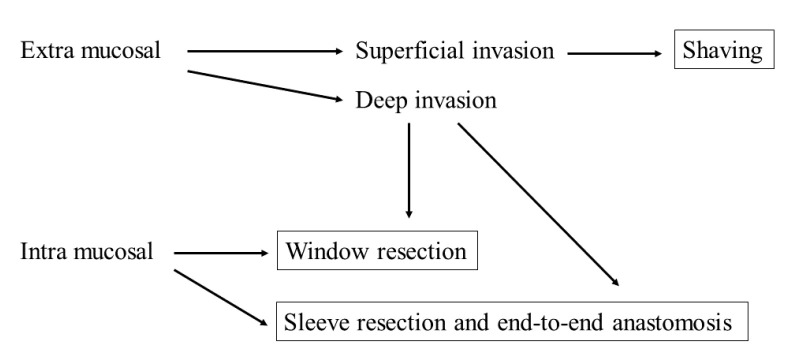
A schema depicting the surgical strategy for tracheal invasion of well-differentiated thyroid cancer according to the extent of invasion.

**Table 1 cancers-13-00797-t001:** Features of the surgical method.

Techniques	Advantage	Disadvantage
Shaving	Maintaining the tracheal lumen Low morbidity	Risk of local recurrence
Window resection	Microscopic resection Postoperative maintenance of airway	The need for multiple surgeries
Sleeve resection with end-to-end anastomosis	More radical resection Lumen is covered with tracheal mucosa	Risk of fatal complications

## Data Availability

No new data were created or analyzed in this study.

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
