# Peer review of "Surgical Management of Tracheal Invasion by Well-Differentiated Thyroid Cancer"

_cancers, 2021, doi:10.3390/cancers13040797_

Round 1

Reviewer 1 Report

Cancers

Review

 Surgical Management of the Tracheal Invasion of Well-differentiated Thyroid Cancer

The authors review the prognostic effects of tracheal invasion and different types of tracheal resection for locally invasive differentiated thyroid carcinoma.

First, the English needs to be meticulously reviewed. The “the” in the title needs to be removed and “of“ changed to “from“ or “by“ for example. It should read :

 Surgical Management of Tracheal Invasion from Well-differentiated Thyroid Cancer.

Line 50: should read “laryngo-tracheal invasion”

Line 63: should read “tracheal invasion by WDTC”

Line 65: “indicated”

Line 67: “but did not specify”

Line 76: “Anatomy of the trachea”

Line 80: “with a membranous portion posteriorly;” delete “and”

Line 89: I would suggest changing “supreme” to “superior”

Line 95: “electrocauterization” – rather say “electrocauterization” if there is a small residual remnant on the trachea.

Line 121: “evinced” is not the correct word here

Etc.

The authors make several statements that aren’t really substantiated by the literature. These are rare tumors and high-level evidence is lacking, and that is one point that the authors need to make very clear. They also need to state that the resection really depends on the extent of the disease. It would be interesting to have a chapter on preoperative work-up for ascertaining the extent of tracheal invasion. The chapter (chapter 4) about endotracheal intubation in not really relevant and the remark about performing a tracheostomy preoperatively is not really substantiated by any published studies.

In the abstract, it is stated that “the trachea is frequently affected in thyroid cancer” but actually it is not the most frequent site of invasion in locally invasive cancer (McCaffrey et al Laryngoscope 2006;111:1-11, reference 3) and is really not “frequent” if you consider all of the thyroid cancers that we encounter, as the data cited shows.

Line 108: frozen section “is recommended”: that is not in any guideline that I am aware of

They state that intraluminal invasion is worse than superficial invasion, but really all we have is data on different types of tracheal resection; there is little to no data about this. Prognosis also depends on esophageal invasion, pathology and distant metastases, which the authors do not really explain.

On line 188, the authors state that “primary closure is rarely possible” but this really depends on the extent of tracheal involvement and tracheal resection. In my experience (Hartl DM et al Head Neck 2013;36:1034-8) and that of Tsukahara et al, primary closure is an option for a significant proportion of patients.

In fact, they only briefly cite Tsukahara K et al (Acta Otolaryngol 2009;129:575-9) who report a series of patients who were treated with a “wedge” or “window” or “segmental” resection preserving the posterior membranous portion of the trachea with primary closure of the trachea and without creation of a tracheo-cutaneous fistula. This is a very important technique when applicable

On line 239 the authors say “en bloc resection” with the thyroid should be performed and then in the next line they contradict themselves. En bloc resection is not necessarily important.

On line 252 the authors say that the sutures are often covered with the sternomastoid muscle. That is maybe their personal experience but it is not frequent to use this muscle in my experience (the sternohyoid muscle may also often be spared and employed).

Line 287: use of monofilament for the tracheal anastomosis is not necessarily the practice of everyone, particularly for the cartilaginous portion of the trachea.

Lines 290-291: the authors go into the debate over chin sutures; there is no evidence from the literature that this practice is useful and actually it’s painful and “barbaric” and has never been shown to decrease the rate of suture dehiscence. This practice has been abandoned in France for at least 25 years now. It think the whole debate is irrelevant.

Reviewer 2 Report

In this paper, the authors reported an interesting synopsis about the surgical management of tracheal invasion in well differentiated thyroid cancer (WDTC).

The surgical techniques are well reported and discussed, however several cited papers are quite old and some sentences should be revised. This could led to a bias in reflecting the real prevalence of tracheal invasion in WDTC in the real clinical life.

Comments

Introduction section:

Too many sentences are without citations.

Lines 35-38 - Please add the citations to these sentences

Line 36 - I suggest to replace the term “typically”. The presence of extrathyroidal progression of a WDTC is a quite rare event, particularly when we talk about of trachea or esophagus invasion.

Line 38 - Trachea is not frequently affected in thyroid cancer, as correctly reported below. Please revise.

Lines 40-41 - The authors cited quite old references (1993-1996), in which high rate of extrathyroidal infiltration in selected patients with primary or recurrence of differentiated thyroid cancer were reported. I suggest to evaluate the possibility to use the following reference

PMID: 28858687, in which the overall prevalence of tracheal invasion is more adherent to the real clinical practice (0.4-0.7%). Please revise.

Line 42 - The sentence “intraluminal extension occurs in 0.5-1.5% of PTC patients” is misleading. In the cited review of Jimmie Honings, “luminal tumor inside the trachea, the most advanced stage of invasion, was visible on endoscopy in only 0.5% to 1.5% of patients presenting for resection”. This not means that this percentage is for all WDTC patients. Please revise this sentence.

Line 47 - The authors stated about “quality of life and improving prognosis”. In order to report the recent experiences and the news technologies able to help the physicians to improve quality of life of these patients, please revise this section and report a more updated bibliography.

Lines 48-49 - Please add a citation to this sentence.

Lines 49-51 - The sentence “…found that laryngeal tracheal invasion was a significant independent predictor of death in 124 (41%) of 292 PTC patients ..” could be misleading in this context. If the authors would report this sentence, they should specify that these patients do not reflect the clinical population of WDTC that nowadays we faced, in which this high percentage of tracheal invasion (42%) cannot be real. Indeed, in this paper the authors selected different types of patients in different years from 1940 to 1995. Reasonably, in these patients cancers were more advanced, ultrasound was not yet available and the knowledge about the clinical behaviour of WDTC was not precise as now.

Line 55-57 “However, in WDTC, even with tracheal invasion, curative resection is often possible with a small margin of safety, and complete resection can ensure a good prognosis “. Please, replace small with large.

Line 111 - This reference talks about circumferential sleeve resection followed by end-to-end anastomosis, but don’t mention the use of intraoperative frozen section. Please revise.

Line 141 - Please specify the reference number for Ito et al.

Line 163 - “shaving ... great”. Please add a verb between shaving and great.

Line 196 - The paper you cited by Ito et al., discussed about the outcome of shaving or laminated resection of the tracheal cartilage in patient with WDTC, and not about tracheal window resection. Please revise.

Line 293 - Please replace AT with At

Line 342 - Please change “treatment” in “treatment radicality”

Conclusions section:

Please report in the conclusion section that for the patients in which a tracheal invasion was suspected, an accurate preoperative workup was a key point in deciding the right treatment to apply. At least CT scan, bronchoscopy and oesophagogastroscopy, are mandatory in this kind of patients.

Moreover, if tracheal invasion was diagnosed before surgery, several minimally-invasive procedures can be performed. Also if it is not a purpose of this review, please report this option in the conclusions.

Reviewer 3 Report

The authors present a review on surgical management of tracheal invasion in thyroid well-differentiated carcinomas. The tracheal involvement is far more often in aggressive thyroid carcinomas, i.e. anaplastic and poorly differentiated carcinomas. This issue should be also discussed.

Recently TNM classification was updated and only macroscopic invasion is valid for T3. Can authors discuss this issue?

The paper would benefit from mind map on surgical decision.

In addition illustration/photographs from surgery would be beneficial for readers.

Reviewer 4 Report

In this review paper the authors present on the management of tracheal invasion of thyroid cancer.

This is a well organized paper, however some of the language phrasing is unconventional and a little hard to follow due to sentence structure and word choices, I would recommend this paper is edited by a professional academic service to improve the understanding for the reader.

For table one it would be helpful to have spaces between the text for each technique or perhaps dotted lines as it is not clear for the advantage column which points correspond to each technique.

For section 3, it would be helpful if each technique was discussed with regards to description and indications, complications, local recurrence and overall survival which could be further divided into adjuvant radiotherapy on local recurrence and overall survival. Currently this information is included in the text but the structure could be improved for readability.

Round 2

Reviewer 1 Report

The authors have extensively revised the manuscript, removing the unsubstantiated parts regarding airway management, adding a review or preoperative work-up  and rewriting the English.

The manuscript has been greatly improved.

In the abstract, the 1st sentence should be modified: instead of “and is characterized by,” replace with “ but may show.”

The sentence regarding MRI as “not as easy to perform as CT or US” is  strange: it may be longer for the patient to stay in the machine, but it’s just as “easy” to perform as CT.

“BS” should be replaced by Bronchoscopy or Direct Tracheoscopy in the subheading  and in the text to be easier to read (BS means something else in English).

The chapter full layer resection needs to be re-organized. There is

1/window resection or full thickness resection preserving the posterior tracheal wall (and thus the vascularization) with primary direct closure and no tracheostomy : mostly for small defects

2/ window resection with tracheostomy or tracheocutaneous fistula : also mostly for small defects, otherwise may require several operations to obtain closure of the fistula

3/ sleeve resection (cartilaginous and membranous resection) with end-to-end anastomosis, with the complications that the authors describe well

4/ other techniques for reconstruction of large tracheal defects.

Reviewer 2 Report

The authors answered to most of questions of my revision.

Reviewer 3 Report

The authors put reasonable arguments and answers to my comments. The paper is improved. 
